# Immune Checkpoint Inhibitor-Induced Myositis/Myocarditis with Myasthenia Gravis-like Misleading Presentation: A Case Series in Intensive Care Unit

**DOI:** 10.3390/jcm11195611

**Published:** 2022-09-23

**Authors:** François Deharo, Julien Carvelli, Jennifer Cautela, Maxime Garcia, Claire Sarles, Andre Maues de Paula, Jérémy Bourenne, Marc Gainnier, Amandine Bichon

**Affiliations:** Marseille Public University Hospital System, 13005 Marseille, France

**Keywords:** myositis, myocarditis, immunotherapy, melanoma, checkpoint inhibitor, immune adverse event, intensive care, myasthenia gravis

## Abstract

Introduction: Immune checkpoint inhibitors (ICIs) are a major breakthrough in cancer treatment. Their increasingly frequent use leads to an uprising incidence of immune-related adverse events (irAEs). Among those, myocarditis is the most reported fatal cardiovascular irAE, frequently associated with ICI-related myositis. Case series: Here, we report three cases of ICI-induced myocarditis/myositis with an extremely severe myasthenia gravis-like (MG-like) presentation, highlighting the main challenges in irAEs management. These patients were over 60 years old and presented an ongoing melanoma, either locally advanced or metastatic, treated with ICI combinations. Shortly after the first or second ICI infusion, they were admitted in an intensive care unit (ICU) for grade 3 ICI-induced MG-like symptoms leading to acute respiratory failure (ARF) requiring invasive mechanical ventilation (IMV). The initial misdiagnosis was later corrected to severe ICI-induced seronegative myocarditis/myositis upon biological results and histopathology from muscular/endomyocardial biopsies. All of them received urgent high-dose corticosteroids pulses. The oldest patient died prematurely, but the two others received targeted therapies leading to complete recovery for one of them. Discussion: These cases highlight the four main challenges of irAEs, encompassing the lack of knowledge among physicians, the risk of misdiagnosis due to numerous and non-specific symptoms, the frequent overlapping forms of irAEs, and the extremely rare MG-like misleading presentation of myocarditis/myositis. The exact pathophysiology of irAEs remains unclear, although a major involvement of the lymphoid compartment (specifically T lymphocytes) was evidenced. Therapeutic management is based on urgent high-dose corticosteroids. For the severest forms of irAEs, case-by-case targeted immunosuppressive therapies should be urgently administered upon multidisciplinary meetings. Conclusion: These cases highlight the lack of knowledge of irAEs among physicians, aggravated by misleading overlapping forms, requiring specific management in trained units and multidisciplinary care. Severe MG-like presentation of irAEs constitutes an absolute therapeutic emergency with high-dose corticosteroids and targeted immunosuppressive therapy.

## 1. Introduction

The recent advent of immune therapies, with checkpoint inhibitors (ICIs) leading the way, was a breakthrough for the treatment of cancers [1]. They aim to generate and reactivate an immune response against cancer [2]. They include programmed cell death 1 (PD-1) inhibitors or their ligand, programmed cell death ligand 1 (PD-L1) inhibitors, cytotoxic T-lymphocyte antigen 4 (CTLA-4) inhibitors [3,4,5], and anti-LAG-3 (Lymphocyte activation gene 3) [6]. Other ICIs are under investigation (phase 1, 2, and 3 trials) and do not have any FDA approval yet (T cell immunoglobulin and mucin-domain containing-3 (TIM-3), T cell immunoglobulin and ITIM domain (TIGIT), B and T lymphocyte attenuator (BTLA), V-domain immunoglobulin suppressor of T cell activation (VISTA) (PD-1H)) [2]. As this new therapeutic strategy was associated with greater outcomes in oncology [1,7,8,9], its use is constantly increasing [1]. However, ICIs can cause potentially fatal immune-related adverse events (irAEs) affecting all organs [10,11]. These irAEs usually occur within the first 2 months after ICI infusions [12]. Muscular and cardiovascular side effects such as myocarditis and myositis are increasingly reported [13,14].

Myocarditis is the most common and fatal cardiovascular toxicity induced by ICIs, reported in fewer than 1% of patients [15], leading to death in 50% of cases [10], and occurring early after the ICI therapy onset, generally within 3 months [14,16]. A frequent association with ICI-related myositis is reported in up to 25% of cases [10].

Current therapies for these adverse events (AE) are based on expert consensus and pathophysiology. The latter mainly involves ICI-induced T lymphocyte hyperactivation and a subsidiary role of common antigens on both cardiac and skeletal muscles. They include an urgent administration of high-dose corticosteroids as first-line therapy (1000 mg/day) [14,17]. Additional immune-modulating therapies are recommended for refractory/severe cases and for specific irAEs. However, these treatments are extrapolated by analogy with similar diseases (for instance, anti-TNFα in colitis, plasma exchange in neurological toxicities, etc.) and remain controversial because they are not based on histopathology of irAEs [18,19,20,21,22,23,24].

Due to their relative novelty, these irAEs are still underestimated and underdiagnosed by practitioners, thus worsening prognosis. Additionally, clinical presentation can be misleading as symptoms can overlap. 

Here, we report three patients with ongoing melanoma admitted in the ICU for grade 3 ICI-induced myocarditis / myositis with a misleading MG-like presentation complicated with ARF requiring IMV. These cases highlight the main challenges of irAEs diagnosis as well as their urgent treatment. 

## 2. Materials and Methods

Adult patients treated with ICIs and admitted in the ICU of the Timone University Hospital (Marseille, France) between 1 January 2016 and 30 June 2021 were retrospectively screened using hospital report-based coding according to the ICD-10. This retrospective, monocentric, and non-interventional study was communicated to the Commission on Data Processing and Freedom and approved by the APHM ethics committee (#2022-20) [25]. Thus, 49 patients admitted in the ICU and presenting myocarditis (I40, I51), myositis (M60), and/or myasthenia gravis (G70) as the principal diagnosis (i.e., the health problem that justified admission to hospital), the related diagnosis (i.e., potential chronic disease or health state during the hospital stay), or a significantly associated diagnosis (i.e., comorbidity or associated complication) were screened. Only 3 of them received ICIs and were included in our case series. The retrospective selection bias was corrected crossing hospital report-based ICD-10 coding search with extraction of patients who received either abatacept and/or plasmapheresis in the ICU during the same period. Data were collected using hospital reports, lab results, and imaging reports.

## 3. Observation

### 3.1. Patient 1

A 65-year-old female with a history of hypertension, obesity (BMI of 35 kg/m²), diabetes mellitus, spontaneously resolving drug-induced pancreatitis with spontaneous resolution, and locally advanced melanoma diagnosed in 2020 was admitted a year later in the intensive care unit (ICU) for dyspnea associated with myasthenia gravis-like symptoms 21 days after the first infusion of anti-PD-1 (nivolumab) and anti-LAG-3 (relatlimab). 

Oncological history reported a five-centimeter subcutaneous nodule located on the left thigh. This nodule was surgically removed, and anatomopathological analyses confirmed the nosology of melanoma harboring activating BRAF V600E mutation, with a resection in healthy margins and the presence of vascular emboli. Revision surgery was required upon multiple melanomas in transit metastases of the left inguinal area enhancing the same anatomopathological features as previously described. Evolution towards non-resectable and locally extended melanoma on the left thigh was treated with the ICI combination of Nivolumab + Relatlimab (BMS020 protocol).

Upon admission in the ICU, the patient reported a symptom onset 16 days after the administration of ICIs. Asthenia was the *primum movens* rapidly evolving towards a florid symptomatology associating major fatigue, ascendant muscular and general weakness without myalgia, right then bilateral ptosis that did not fluctuate over time, binocular diplopia and ultimately stage III dyspnea, and dysphonia. At the ICU arrival, myasthenia gravis (MG)-like symptoms were noted with a quantitative MG score of 60/100. [26] Twelve-lead electrocardiogram showed a new right bundle branch block pattern without other abnormality. Transthoracic echocardiogram (TTE) was normal. Biological findings revealed an elevation of troponin at 670 ng/L (normal range 0–14) and creatine phosphokinase (CPK) at 7397 UI/L, (normal range: 0–190). Inflammatory biomarkers were normal. A baseline dNLR of 3.25 and an elevated NLR at 14.6 were assessed. Thyroid hormone levels were normal. The immunologic screening ruled out a potential underlying autoimmune disease with normal complement levels, unspecific speckled antinuclear antibodies with a low titer of 1/160, absence of rheumatoid factor, anti-citrullinated peptide antibody, soluble antigen or autoimmune hepatitis-related antibodies, and normal immunoglobulin quantitation. Additionally, both antibodies to acetylcholine receptor (AChR) and anti-muscle specific kinase (MuSK) were negative. Paraclinical investigations (body computed tomography scan (body CT) and ENMG) discarded neurological diseases, specifically multiple sclerosis, MG, demyelinating polyradiculoneuropathy, and other peripheral neuropathies.

A muscular biopsy of right quadriceps was performed, revealing signs of denervation with atrophic fibers, some even reduced to nuclear bags (Figure 1a). Discrete edema was encountered within the interstitial space, although no inflammatory infiltrate was observed. Histoenzymatic reactions were normal. Amyloses was discarded. A dysimmune profile was reported by immune-histo-chemical analysis. This interpretation might have been hindered by the prior administration of irAEs-targeted treatments. 

Upon the suspected diagnosis of ICI-induced myocarditis associated with myositis, the patient received urgent intravenous corticosteroid course (1 mg/kg/day). Her condition secondarily deteriorated, with ARF requiring IMV and rapid ventricular arrhythmia requiring emergent electrical cardioversion. Although constantly increasing troponin levels were observed with a maximum level of 1055 ng/L, cardiac ultrasound close monitoring remained normal. Cardiac magnetic resonance imaging (cardiac MRI) showed signs of myocarditis with T2 myocardial signal enhancement overlying anterior and apical left ventricle wall. The coronary angiography was normal (a few parietal irregularities on left anterior descending artery <30% were considered insignificant and left untreated). Confronted with refractory ICI-induced myocarditis associated with myositis, specific therapies were initiated with high-dose methylprednisolone (1000 mg/day for 3 days, then 2 mg/kg/day), repeated plasmapheresis (seven sessions over the ICU stay), and abatacept infusions (500 mg at day 0 and day 15). 

The overall outcome was favorable as the patient was discharged from the hospital fifty days after the onset of irAEs. Myositis fully and rapidly resolved but ascending troponin levels concomitant to corticosteroid tapering required a slow weaning. The follow-up at six months after the irAEs diagnosis revealed an ongoing oral prednisolone (10 mg/day) intake, persistently elevated troponin levels (110 ng/L) with normal ECG and TTE, normal CPK levels, and a locoregional progression of the melanoma treated with targeted therapy (encorafenib/binimetinib). Nine months after ICU admission, partial response of metastatic melanoma was observed, and an almost complete weaning of steroids was achieved as the patient received sole hydrocortisone 20 mg/day to prevent acute adrenal insufficiency (troponin 44 ng/L and unchanged TTE). A year after the myocarditis, the patient showed complete response of melanoma, and troponin levels were normal even after complete weaning of corticosteroids. Figure 2a. illustrates the chronology of events (symptoms, biomarkers, and treatments).

### 3.2. Patient 2

An 84-year-old male with a history of hypertension, asthma, and metastatic melanoma diagnosed in 2019 was admitted in the ICU for dyspnea associated with MG-like symptoms 19 days after the first infusion of combined anti-PD-1 (nivolumab) and anti-CTLA-4 (ipilimumab). 

An oral mucosa malignant melanoma BRAF^V600E^- was diagnosed in the biopsy of a twenty-centimeter-large tumor of the left nasal cavity. The unresectable and metastatic (pancreas) character of the melanoma led to an ICI combination of Nivolumab + Ipilimumab as the first-line treatment upon validation by the onco-geriatric team. 

Upon admission in the ICU, the patient reported a symptom onset 12 days after the first infusion of ICIs. Initial general muscular weakness rapidly evolved towards a florid symptomatology associating myalgia, left then bilateral ptosis that did not fluctuate over time, binocular diplopia and, ultimately, stage IV dyspnea, dysphagia, and dysphonia. Upon arrival in the ICU, the patient also presented with a ubiquitous maculopapular rash. A twelve-lead electrocardiogram showed a left bundle branch block pattern and high-degree atrioventricular block. According to the electro-physiologist’s advice, temporary right ventricular pacing was postponed. TTE revealed no abnormality, particularly no change in ventricular function or segmental kinetic disorder. Biological screening showed elevated troponin at 2371 ng/L (normal range 0–14) and CPK levels at 7683 UI/L, (normal range: 0–190). Inflammatory biomarkers revealed an elevated CRP at 63 mg/L, (normal range: 0–5). A baseline dNLR of 18 and an elevated NLR at 27 were assessed. Elevated liver enzymes were noted at 743 UI/L and 602 UI/L for ALAT and ASAT, respectively, without viral or autoimmune component. Thyroid hormone levels were normal. Immunologically, complement levels were normal, along with plasma protein electrophoresis and normal immunoglobulin quantitation. Lone and unspecific speckled antinuclear antibodies at a low titer of 1/160 and a rheumatoid factor at 36 mg/L (normal range: 0–20) were encountered. Both antibodies for AChR and MuSK were absent. 

A muscular biopsy of left deltoid was performed (Figure 1b), discarding amyloses but revealing signs of dermatomyositis with irregular and atrophic muscle fibers, some even reduced to nuclear bags. A major interstitial inflammatory infiltrate of macrophages and mononuclear lymphocytes was described along with fibers in necrosis regeneration. Diffuse MHC I antigen expression was enhanced in perifascicular and inflammatory areas. 

The patient rapidly deteriorated, with ARF requiring IMV the day after ICU admission. Both cardiac MRI and coronary angiography were postponed due to respiratory and hemodynamic instability. Upon a high probability of ICI-induced acute myocarditis associated with myositis and hepatitis, the patient received urgent high-dose corticosteroid pulses (1000 mg/day for 3 days, followed by 2 mg/kg/day). 

Despite the corticosteroid therapy and decreased cytolysis and CK levels, the evolution was unfavorable, and the patient died before planned plasmapheresis at day 5 after admission in the ICU. Figure 2b illustrates the chronology of events (symptoms, biomarkers, and treatments). 

### 3.3. Patient 3

A 70-year-old male with a history of hypertension, dyslipidemia, diabetes mellitus, and metastatic melanoma diagnosed in 2019 was admitted in the ICU for an asymptomatic elevated troponin 30 days after the second infusion of combined anti-PD-1 (nivolumab), anti-CTLA-4 (ipilimumab), and anti-LAG-3 (relatlimab) (BMS CA224-048). 

Stage IV superficial spreading melanoma (SSM) BRAF^V600E^ + was diagnosed upon behavioral disorders and aphasia, revealing a cerebral lesion on the left frontal lobe. Surgical removal of the latter evidenced a melanoma metastasis. A subcutaneous nodule on the left shoulder was further identified as the primary lesion. First-line treatment provided an ICI combination (BMS048 protocol).

After the first ICI infusion, the patient presented twice in the emergency ward for febrile tachycardia attributed to ICIs at day 1 and 3 after the infusion, respectively. No specific treatment was administered, and the patient was discharged. At day fifteen, he received the second combination of ICIs. A fortuitous and asymptomatic elevated troponin at 280 ng/L (normal < 14) in planned blood analyses led to the admission in cardiology two weeks after the second infusion. At arrival, the patient was still asymptomatic, and the twelve-lead electrocardiogram showed sinus tachycardia and a new right bundle branch block pattern. TTE revealed lone right ventricular dilatation. Biological findings showed elevated levels of troponin (293 ng/L), CPK (2240 UI/L), and liver enzymes (120 UI/L). Although chest CT scan and coronary angiography were normal, the cardiac MRI showed signs of myocarditis involving both ventricles with late gadolinium enhancement and regional myocardial signal increase in T2 with inferior, anterior, and apical distribution. The endomyocardial biopsy enhanced features of iatrogenic myocarditis with inflammatory infiltrates of mononuclear lymphocytes and macrophages along with cardiomyocyte necrosis (Figure 1c). Thus, high-dose methylprednisolone (1000 mg/day pulses for three days, then 2 mg/kg/day) was administered at day 19 after second ICI combination. In addition to systemic corticosteroids, additional specific therapy with infliximab was administered (500 mg) twenty-three days after the second ICI infusion due to ascending troponin levels (869 ng/L). An intercurrent episode of high-degree atrioventricular (AV) block was diagnosed at day 26 and treated with the implementation of a permanent endocavitary pacemaker.

The patient was transferred to neurology at day 32 and in the ICU at day 38 due to MG-like symptoms associating binocular diplopia, bilateral ptosis, proximal muscular weakness, dysphonia, and a myasthenic score of 60/100. Concomitant serum levels of CPK at 818 UI/L and troponin at 695 ng/L were noted. Targeted treatment included IVIG 0.4 g/kg/day for five days and prostigmine. Inflammatory biomarkers were normal. A baseline dNLR of 2 and an elevated NLR of 3.6 were assessed. Thyroid hormone levels were normal. The immunologic screening ruled out a potential underlying autoimmune disease with normal complement levels, unspecific speckled antinuclear antibodies with a low titer of 1/160, absence of rheumatoid factor, anti-citrullinated peptide antibody, soluble antigen or autoimmune hepatitis-related antibodies, and normal immunoglobulin quantitation. Additionally, both antibodies for acetylcholine receptor (AChR) and anti-muscle specific kinase (MuSK) were negative. Paraclinical investigations (body computed tomography scan (body CT) and ENMG) discarded neurological diseases, specifically multiple sclerosis, MG, demyelinating polyradiculoneuropathy, and other peripheral neuropathies. 

His condition secondarily deteriorated, with ARF requiring IMV and rapid ventricular arrhythmia requiring emergent electrical cardioversion. Confronted with refractory ICI-induced myocarditis/myositis/MG, specific therapies were initiated with abatacept (500 mg) at day 48 and repeated plasmapheresis (five sessions) from day 48 to 60.

Despite specific treatments improving hemodynamic and neurological status, and decreasing CK and troponin levels at 64 UI/L and 822 ng/L, respectively, the patient died 62 days after the second infusion of ICI combination. Figure 2c. illustrates the chronology of events (symptoms, biomarkers, and treatments). 

## 4. Discussion

irAEs are an increasingly studied and reported entity in the oncological field [12,27]. Our case series highlight the four main challenges regarding irAEs, encompassing: -The lack of knowledge among physicians, thus delaying diagnosis and underestimating the severity [28].-The frequent overlapping forms of irAEs [12,29], with confusing clinical presentation rendering difficult the accurate diagnosis (Figure 3) for emergency treatment.-MG-like misleading presentation of myocarditis/myositis. Although the association of ICI-induced MG/myositis was described [29], only five cases of ICI-induced myositis initially presenting as misleading MG-like were reported in the literature prior to our series [30].

**Figure 3 jcm-11-05611-f003:**
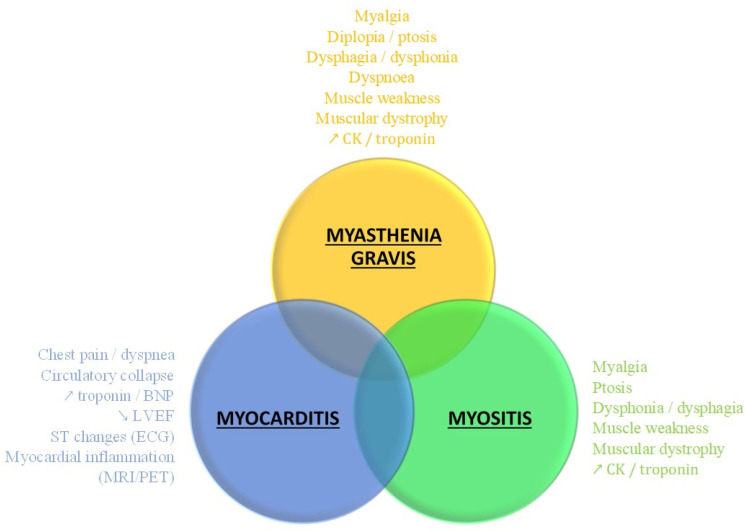
Overlapping forms of severe irAEs. irAE: immune-related adverse event; CPK: creatin phosphokinase; BNP: bone natriuretic peptide; LVEF: left ventricular ejection fraction; ECG: electrocardiogram; MRI: magnetic resonance imaging; PET: positron emission tomography.

Thus, the diffusion of basic knowledge and irAEs’ standard of care among emergency physicians and intensivists constitutes the key to early and accurate diagnosis, proper orientation, and urgent targeted treatment as both overall prognosis and irAEs’ reversibility improve with early care [31,32]. Prognosis also depends on the grade of irAEs and on the association between irAEs [12].

Alongside the aforementioned acknowledgement of irAEs and standardized protocols, an early assessment of risk factors for severe irAEs should be performed beforehand. [33,34] risk factors are clinical (younger age [35], BMI >23 kg/m² [31], current or former smoking status [36], history of type I hypersensitivity [37]), cancer-related [35], drug-related (multiple cycles of pembrolizumab [31], the use of PD1 or CTLA4 inhibitors [36], and ICI combination [35]) and biological [36].

Pathophysiology underlying irAEs likely involves multiple mechanisms [2]. Involvement of the lymphoid compartment with ICI-induced T cell hyperactivation is a cornerstone of the pathophysiology of irAEs. Indeed, tissue infiltration with activated T lymphocytes has been demonstrated [38], notably with CD8+ T cell infiltrates in the myocardial tissue [13]. The muscular/endomyocardial biopsies performed in our case series also revealed significant CD3+, CD4+, and CD8+ lymphocyte infiltrates (Figure 1a–c). ICIs impact immunologic homeostasis by changing the T cell repertoire, thus leading to a self-reactive T cell contingent (above with antiCTLA4 use), cytokine-producing T cells [39], and clonal expansion of rearranged TCR V-b sequences [40]. ICIs also interfere with self-tolerance by depleting regulatory T cells, thereby activating previously anergic self-reactive T cells [41]. Shared TCR clonality between tumor-infiltrating and myocardial-infiltrating T cells without IgG deposits may explain the occurrence of ICI-related myocarditis. [12], ICIs also prevent inhibitory signaling pathways leading to an overexpression of allergen-specific CD4+ T cells involved in Th2 differentiation, cytokine release (IFN-γ, TNF-α, IL-5, IL-13 or IL-17F, IL-4), and immunoglobulin class switching of B cells, thus promoting type I hypersensitivity reactions and irAEs [37]. Other pathophysiological mechanisms have been hypothesized, though displaying a minor or inconsistent role in the development of irAEs (preexisting autoimmune conditions [42,43] with circulating autoantibodies [38,43], shared self-antigen expression by both healthy tissues and the tumor [44], myeloid compartment disturbances with increased levels of circulating plasmablasts, increased cytokine production [20,38], and direct toxicity of ICIs binding to both tissues and complement [38]). irAEs are a double-edged sword, as their occurrence is a positive predictive factor of treatment response [42,43,45], while grade 3/4 irAEs can be fatal. In regard to pathophysiological findings highlighting a major role of hyperactivated T lymphocytes, immunosuppressive therapies targeting T cells rather than antibody depletion (plasmapheresis, IVIG) seem the best option for irAEs. 

Although potentially life-threatening, irAEs have been significantly associated with an ICI benefit regarding overall response rate, progression-free survival, and overall survival in patients with recurrent/metastatic solid cancer [46,47,48]. Our first case illustrates this hypothesis.

Regarding treatment of irAEs, recent American and European guidelines recommended to stop ICIs and to urgently start high-dose corticosteroids (1–2 mg/kg/day), preferably within the first 24 h following symptoms [10,17]. In addition to immediate care, case 1 also highlights the necessity of a very progressive and careful weaning over several. Indeed, steroid tapering should be considered after 48 h of consistent symptom improvement and extended over 4–6 weeks [12]. A scrupulous and large monitoring is recommended, as troponin levels alone were not systematically correlated with the risk of cardiovascular events [49]. 

Immunosuppressive therapies (infliximab, mycophenolate mofetil, antithymocyte globulin, and tacrolimus) are generally proposed for refractory and severe forms of irAEs [10,50]. Nonetheless, ICU patients diagnosed with ICI-induced myositis/myocarditis should be urgently treated with a T-cell-targeted therapy concurrently with steroids as it conditions the prognosis. The choice of treatment targeting irAEs is driven by the aforementioned histopathology [20] and the organ-specific lesion. Abatacept seem the most accurate therapy for ICI-induced myocarditis [13,51]. 

We propose in Figure 4. a brief summary to promptly detect patients with irAEs in the emergency or intensive care wards, along with an initial screening and accurate care within the first 24 h.

This paper has some limits as we described solely three cases. Additionally, histopathology was suboptimal, as late muscular biopsy was performed in case 1, possibly concealing inflammatory signs, and endomyocardial biopsy was not performed for this patient. The strengths of this article include the rarity and peculiarity of the clinical presentation, the highlights on the need for better diffusion of irAEs acknowledgment among physicians, and the long-lasting follow-up of nine months for the surviving patient.

## 5. Conclusions

We described three cases of ICI-induced severe myocarditis/myositis with a misleading MG-like presentation. These cases firstly highlight the necessity to manage these patients in specified units and, secondly, the absolute urgency of initiating aggressive immunosuppressive treatment with high-dose corticosteroids combined with anti-T cell treatment chosen upon multidisciplinary decision.

## Figures and Tables

**Figure 1 jcm-11-05611-f001:**
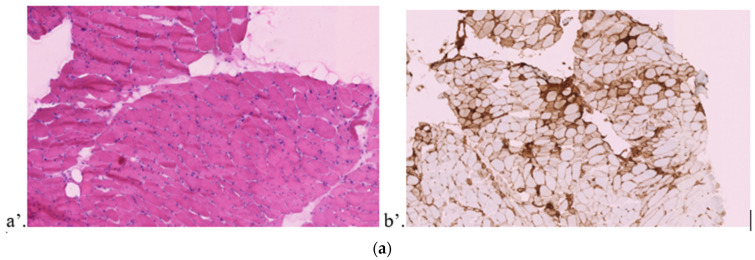
Muscular/endomyocardial biopsies. (**a**) Right quadriceps muscular biopsy of case 1. (**a’**) Denervation: atrophic fibers and nuclear bags. (**b’**) Diffuse expression of MHC I. (**b**) Left deltoid muscular biopsy of case 2. (**a’**) Perifascicular atrophy, necrosis, and inflammation. (**b’**) Diffuse expression of MHC I with perifascicular and peri-inflammatory enhancement. (**c’**) CD3+ lymphocytes in the inflammatory infiltrate. (**d’**) CD4+ lymphocytes in the inflammatory infiltrate. (**e’**) CD68+ in the inflammatory infiltrate. (**f’**) PD1+ lymphocyte expression in the inflammatory infiltrate. (**c**) Endomyocardial biopsy of case 3. (**a’**) Myocarditis with cardiomyocyte necrosis compatible with a toxic drug origin. (**b’**) Mononuclear macrophages (CD68+) and lymphocytes (CD8+ and CD4+) in inflammatory infiltrates. Macrophages (CD68+) partially penetrate necrotic cardiomyocytes.

**Figure 2 jcm-11-05611-f002:**
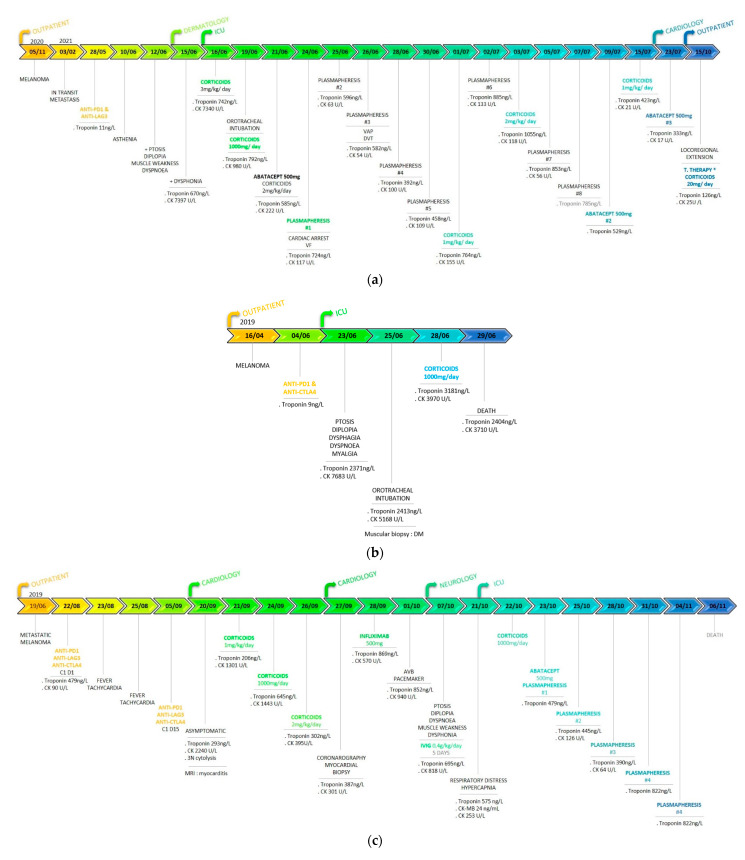
Chronology of events (diagnosis, symptoms, biomarkers, and treatments). (**a**) Patient 1. CK: creatine kinase; VF: ventricular fibrillation; VAP: ventilator-acquired pneumonia; DVT: deep-vein thrombosis.*Targeted therapy: encorafenib and binimetinib. (**b**) Patient 2. CK: creatin kinase; DM: dermatomyositis. (**c**) Patient 3. C1D1: first day of the first cure; CK: creatine kinase; MRI: magnetic resonance imaging; AVB: atrioventricular block. (**d**) Comparative table of the three reported patients.

**Figure 4 jcm-11-05611-f004:**
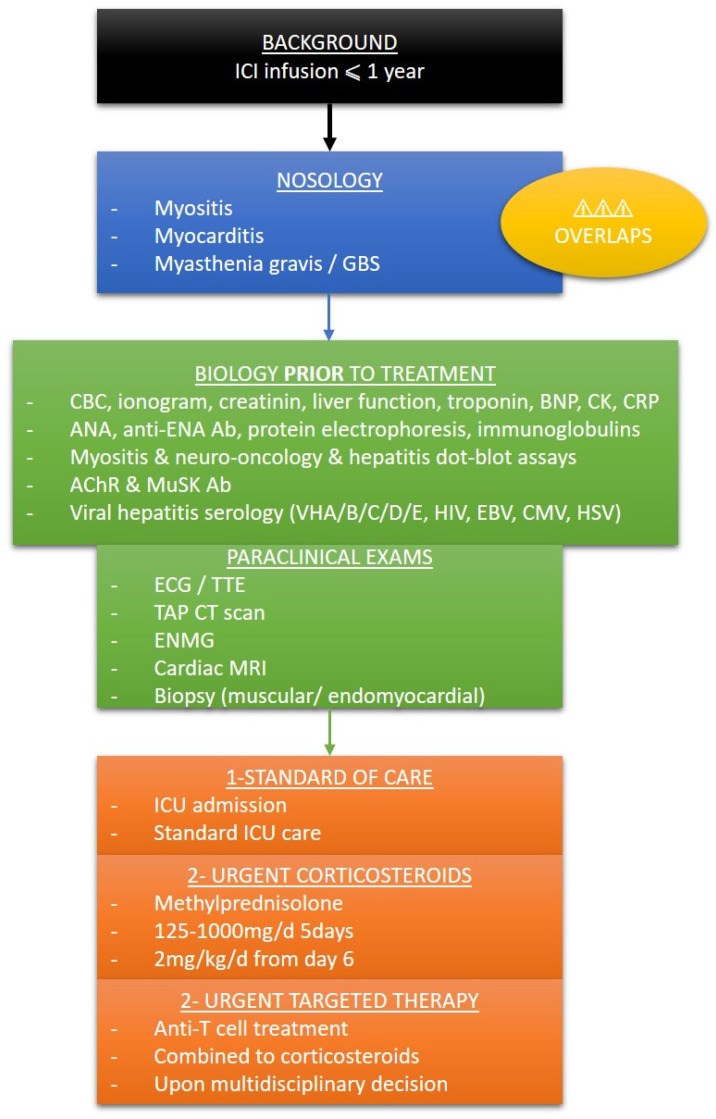
Simplified protocol for screening and care of irAEs in the emergency ward or ICU. ICI: immune checkpoint inhibitor; GBS: Guillain–Barré syndrome; CBC: complete blood count; BNP: bone natriuretic peptide; CK: creatin kinase; CRP: C reactive protein; Ab: antibodies; ANA: antinuclear antibodies; anti ENA: extractable nuclear antigen; AChR: acetylcholine receptor; MuSK: muscle specific kinase; EBV: Epstein-Barr virus; CMV: cytomegalovirus; HSV: herpes simplex virus; ECG: electrocardiogram; TTE: transthoracic echocardiogram; TAP CT scan: thoracoabdominopelvic computed tomography scan; ENMG: electroneuromyogram; MRI: magnetic resonance imaging; ICU: intensive care unit.

## Data Availability

The data presented in this study are available on request from the corresponding author.

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
