# Peer review of "Immune Checkpoint Inhibitor-Induced Myositis/Myocarditis with Myasthenia Gravis-like Misleading Presentation: A Case Series in Intensive Care Unit"

_jcm, 2022, doi:10.3390/jcm11195611_

Round 1

Reviewer 1 Report

The article entitled "Immune checkpoint inhibitor-induced myositis/myocarditis with myasthenia gravis-like misleading presentations: a case series in intensive care unit" presents a review of three cases of myocarditis and myositis associated with myasthenia gravis-like, in addition to a review of the literature on immunotoxicity. This article shows us a rare complication of immunotherapy that happens in less than 1% of patients, and that due to its severity it is essential to know its pathophysiology, diagnosis, and treatment. The article made by the authors I think is of great value for its presentation to the scientific community, and from my point of view, it can be accepted for publication if a series of changes are made that are detailed below. The work carried out despite its limitation because it is a series of cases can allow to have a greater knowledge of the irAEs, being fundamental for the management of future patients with these complications, replacing the authors the deficiency of the sample size with a good review of immunotoxicity.

The article is well structured, being easy to read due to the figures provided by the authors that enrich the article. The use of the language is correct. The references used are mostly current and make a good reflection of the main literature on the subject being addressed. The title and keywords are also correct, and I wouldn't change anything about it. The figures are the point of greatest value of the article, being able to make a quick reading of the article through the three figures that summarize the cases of the patients and the algorithm of diagnosis and treatment of this immunotoxicity. One of the points to correct in the article is the lack of a comparative table of the three patients with their clinical differences, symptoms, diagnosis, treatment, and evolution. The following are the proposed changes to the article:

Major changes -As I indicate in the previous section, I think that the most important change that the article needs is the realization of a table that compares the three patients described in the article. -Conclusions: I think it is necessary to reform this part completely. Throughout the article it is specified that non-oncologists do not present adequate training on irAEs, however, I believe that this statement should be withdrawn and affirm that it is necessary to manage these patients in specialized units, regardless of whether the specialists are oncologists or not. It is key to specify that a multidisciplinary management of these patients must be carried out. This point should be changed throughout the entire article.

Minor changes

-To help you read the article, you could add a list of abbreviations.

-Abstract: modify the conclusions section.

-Background, line 40: add anti-LAG3 because its use has already been approved by the FDA for the treatment of metastatic melanoma. 

-Background, line 41: there are other ICIs that are in phase 3, apart from phases 1 and 2. Modify this point

-Background, line 45: indicate that it affects practically all organs due to its effect on the immune system.

-Background, line 55: indicate the recommended doses of corticosteroids. It is important for the reader to know that in these cases the doses indicated are much higher than those of other irAEs.

-Case 1,2 and 3: change the title of this section, instead of Case 1 change to Patient 1.

Case 1, line 85: obesity include as an antecedent, not as a description of the patient.

-Case 1, line 105: define TTE.

Case 1: it is striking that, in a patient with the described history, coronary angiography is normal. Was there no atheromatous plaque or other alteration? Please specify this point if there were any alterations.

Case 2: It is rare to see an 84-year-old patient being treated with the combination of nivolumab-ipilimumab instead of another type of treatment. Specify why this treatment was chosen for this patient.

-Case 3, line 218: indicate that the combination used for this patient is not the standard one. If it was within a clinical trial, please specify.

-Discussion: in the case of the first patient, a complete response of the lesions was observed. Add a paragraph that talks about the association of toxicity and response, because multiple studies indicate that there may be an association between response and immunotoxicity.

-Figure 4: in paraclinical exams cardiac MRI could be added in centers where it is available.

-References: I think the following references should be added that talk about the standard of treatment in metastatic melanoma.

1.Tawbi et al, Relatlimab and Nivolumab versus Nivolumab in Untreated Advanced Melanoma, NEJM 2022.

2.Larkin et al, Combined Nivolumab and Ipilimumab or Monotherapy in Untreated Melanoma, NEJM 2015. 

Author Response

Each comment of the reviewer #1 has been answered accordingly directly in the paper. 

Reviewer 2 Report

I thank the authors very much for this interesting research. The authors provide a case series of three patients presenting myocarditis with severe myasthenia gravis-like symptoms as immune-related adverse events as a result of immune checkpoint inhibitor treatment (ICI). All patients have been treated with an ICI combination because of malignant melanoma in different stages. The irAEs were initially misdiagnosed which leads the authors to the conclusion that among other challenges in diagnosis and treatment of irAEs there is a lack of knowledge among non-specialists in oncology.

After presenting the three cases, the authors provide a protocol for screening and treatment of irAEs at the emergency ward or intensive care unit. 

The strength of this manuscript is that the authors draw attention to very rare irAEs that every physician working with ICIs should be aware of. It is very important to know that a lot of symptoms can overlap and that irAEs can occur even months after treatment. The authors provide a treatment proposal and a flow chart of how to do it in ICU which is very helpful.

One of the weaknesses is the low patient number. Only 49 patients were screened. Maybe it would have been better to have screened the patients with ICI presenting in ICU and not only the patients presenting with myocarditis, myositis, or myasthenia gravis.  

- Please state something or explain in a bit more detail the time frame in which irAEs can occur. This is also very important for (non-)oncologists to know to consider a possible irAE due to ICI given in the past.

- All figures: The size is not ideal, they are a bit pixelated. Please provide more readable font, possibly consider pdf. 

- Fig. 3: Frames are not complete. Font size is very small.

- Fig. 4: try to make it more clear; provide the abbreviation for SOC

- Please correct formal errors (there are a lot of empty spaces etc.)

- Please consider professional english language editing.

Author Response

Thank you for reviewing our paper : 

  • regarding the low number of patients: this limit was mentioned in the discussion section. Retrospective analyses include a selection bias. We purposedly chose the international coding system based on hospital reports and considered a relatively large frametime.
  • all the comments of reviewer #2 have been answered accordingly directly in the paper
  • this article was already professionally edited for english language prior to submission

Case Report

Immune checkpoint inhibitor-induced myositis/myocarditis with myasthenia gravis-like misleading presentation: a case series in intensive care unit

François Deharo, Julien Carvelli, Jennifer Cautela, Maxime Garcia, Claire Sarles, Andre Maues de Paula, Jérémy Bourenne, Marc Gainnier and Amandine Bichon*

Citation: Lastname, F.; Lastname, F.; Lastname, F. Title. J. Clin. Med. 2022, 11, x. https://doi.org/10.3390/xxxxx

Academic Editor: Firstname Lastname

Received: date

Accepted: date

Published: date

Publisher’s Note: MDPI stays neutral with regard to jurisdictional claims in published maps and institutional affiliations.

Copyright: © 2021 by the authors. Submitted for possible open access publication under the terms and conditions of the Creative Commons Attribution (CC BY) license (https://creativecommons.org/licenses/by/4.0/).

    Marseille Public University Hospital System, France

*   Correspondence: amandine.bichon@ap-hm.fr

Abstract: Introduction: Immune checkpoint inhibitors (ICIs) are a major breakthrough in cancer treatment. Their increasingly frequent use leads to an uprising incidence of immune-related adverse events (irAEs). Among those, myocarditis is the most reported fatal cardiovascular irAE, frequently associated with ICI-related myositis. Case series: We report here three cases of ICI-induced myocarditis/myositis with an extremely severe myasthenia gravis-like (MG-like) presentation, highlighting the main challenges in irAEs management. These patients were over 60 years old and presented an ongoing melanoma, either locally advanced or metastatic, treated with ICI combinations. Shortly after the first or second ICI infusion, they were admitted in intensive care unit (ICU) for grade 3 ICI-induced MG-like symptoms leading to acute respiratory failure (ARF) requiring invasive mechanical ventilation (IMV). The initial misdiagnosis was later corrected to severe ICI-induced seronegative myocarditis/myositis upon biological results and histopathology from muscular/endomyocardial biopsies. All of them received urgent high dose corticosteroids pulses. The oldest patient died prematurely but the two others received targeted therapies leading to complete recovery for one of them. Discussion: These cases highlight the four main challenges of irAEs, encompassing the lack of knowledge among physicians, the risk of misdiagnosis due to numerous and non-specific symptoms, the frequent overlapping forms of irAEs, and the extremely rare MG-like misleading presentation of myocarditis/myositis. The exact pathophysiology of irAEs remains unclear, although a major involvement of the lymphoid compartment (specifically T lymphocytes) was evidenced. Therapeutic management is based on urgent high dose corticosteroids. For the severest forms of irAEs, case-by-case targeted immunosuppressive therapies should be urgently administered upon multidisciplinary meetings. Conclusion: These cases highlight the lack of knowledge of irAEs among physicians, aggravated by misleading overlapping forms, requiring a specific management in trained units and a multidisciplinary care. Severe MG-like presentation of irAEs constitute an absolute therapeutic emergency with high dose corticosteroids and targeted immunosuppressive therapy.

Keywords: myositis; myocarditis; immunotherapy; melanoma; checkpoint inhibitor; immune adverse event; intensive care; myasthenia gravis

  1. Introduction

Recent advent of immune therapies, with checkpoint inhibitors (ICIs) leading the way, was a breakthrough for treatment of cancers1. They aim  to generate and reactivate an immune response against cancer2. They include programmed cell death 1 (PD-1) inhibitors or its ligand, programmed cell death ligand 1 (PD-L1) inhibitors, cytotoxic T-lymphocyte antigen 4 (CTLA-4) inhibitors3,4,5 and anti-LAG-3 (Lymphocyte activation gene 3)6. Other ICIs are under investigation (phase 1, 2 and 3 trials) and do not have yet any FDA approval (TIM-3, TIGIT, BTLA, VISTA (PD-1H))2. As this new therapeutic strategy was associated with greater outcomes in oncology1,7–9, its use is constantly increasing1. However, ICIs can cause potentially fatal immune-related adverse events (irAEs) affecting all organs10,11. These irAEs usually occur within the first 2 months after ICI infusions12. Muscular and cardiovascular side effects such as myocarditis and myositis increasingly reported13,14.

Myocarditis is the most common and fatal cardiovascular toxicity induced by ICIs, reported in  less than 1% of patients15, leading to death in 50% of cases10, and occurring early after the ICI therapy onset, generally within 3 months 14,16. A frequent association with ICI-related myositis is reported in up to 25% of cases10.

Current therapies for these adverse events (AE) are based on expert consensus and pathophysiology. The latter mainly involves ICI-induced T lymphocyte hyperactivation and a subsidiary role of common antigens on both cardiac and skeletal muscles. They include an urgent administration of high-dose corticosteroids as first-line therapy (1000mg/day)14,17. Additional immune-modulating therapies are recommended for refractory/severe cases and for specific irAEs. However, these treatments are extrapolated by analogy with similar diseases (for instance, anti-TNFα in colitis, plasma exchange in neurological toxicities etc…) and remain controversial because they are not based on histopathology of irAEs18,19,20,21,22,23,24.

Due to their relative novelty, these irAEs are still underestimated and underdiagnosed by practitioners, thus worsening prognosis. Besides, clinical presentation can be misleading as symptoms can overlap. 

We report here three patients with ongoing melanoma admitted in ICU for grade 3 ICI-induced myocarditis / myositis with a misleading MG-like presentation complicated with ARF requiring IMV. These cases highlight the main challenges of irAEs diagnosis as well as their urgent treatment.

  1. Materials and Methods

Adult patients treated with ICIs and admitted in the ICU of the Timone University Hospital (Marseille, France) between 01/01/2016 and 06/30/2021 were retrospectively screened using hospital report-based coding according to the ICD-10.  This retrospective, monocentric, non-interventional study was communicated to the Commission on Data Processing and Freedom and approved by the APHM ethic committee (#2022-20)25. Thus, 49 patients admitted in ICU and presenting myocarditis (I40, I51), myositis (M60), and/or myasthenia gravis (G70) as the principal diagnosis (i.e., the health problem that justified admission to hospital), the related diagnosis (i.e., potential chronic disease or health state during the hospital stay), or a significantly associated diagnosis (i.e., comorbidity or associated complication) were screened. Only 3 of them received ICIs and were included in our case series. The retrospective selection bias was corrected crossing hospital report-based ICD-10 coding search with extraction of patients who received either abatacept and/or plasmapheresis in ICU during the same period. Data was collected using hospital reports, lab results and imaging reports.

  1. Observation

3.1. Patient 1

A 65 year-old female with a history of hypertension, obesity (BMI of 35 kg/m²), diabetes mellitus, spontaneously resolving drug-induced pancreatitis with spontaneous resolution and locally advanced melanoma diagnosed in 2020, was admitted a year later in intensive care unit (ICU) for dyspnea associated with myasthenia gravis-like symptoms 21 days after the first infusion of anti-PD-1 (nivolumab) and anti-LAG-3 (relatlimab).

Oncological history reported a five-centimeter subcutaneous nodule located on the left thigh. This nodule was surgically removed and anatomopathological analyses confirmed the nosology of melanoma harboring activating BRAF V600E mutation, with a resection in healthy margins and the presence of vascular emboli. Revision surgery was required upon multiple melanoma in transit metastases of the left inguinal area enhancing the same anatomopathological features as previously described. Evolution towards non resectable and locally extended melanoma on the left thigh was treated with the ICI combination of Nivolumab + Relatlimab (BMS020 protocol).

Upon admission in ICU, the patient reported a symptom onset 16 days after the administration of ICIs. Asthenia was the primum movens rapidly evolving towards a florid symptomatology associating major fatigue, ascendant muscular and general weakness without myalgia, right then bilateral ptosis that did not fluctuate over time, binocular diplopia and ultimately stage III dyspnea, and dysphonia. At ICU arrival, myasthenia gravis (MG)-like symptoms were noted with a quantitative MG score of 60/100. 26 Twelve-lead electrocardiogram showed a new right bundle branch block pattern without other abnormality. Transthoracic echocardiogram (TTE) was normal. Biological findings revealed an elevation of troponin at 670 ng/l (normal range 0-14) and creatine phosphokinase (CPK) at 7397 UI/L, (normal range: 0-190). Inflammatory biomarkers were normal. A baseline dNLR of 3.25 and an elevated NLR at 14.6 were assessed.  Thyroid hormone levels were normal. The immunologic screening ruled out a potential underlying autoimmune disease with normal complement levels, unspecific speckled antinuclear antibodies with a low titer of 1/160, absence of rheumatoid factor, anti-citrullinated peptide antibody, soluble antigen or autoimmune hepatitis-related antibodies, and normal immunoglobulin quantitation. Besides, both antibodies to acetylcholine receptor (AChR) and anti-muscle specific kinase (MuSK) were negative. Paraclinical investigations (body computed tomography scan (body CT) and ENMG) discarded neurological diseases, specifically multiple sclerosis, MG, demyelinating polyradiculoneuropathy, and other peripheral neuropathies.

A muscular biopsy of right quadriceps was performed, revealing signs of denervation with atrophic fibers, some even reduced to nuclear bags (Figure 1.a.). Discrete edema was encountered within the interstitial space although no inflammatory infiltrate was observed. Histoenzymatic reactions were normal. Amyloses was discarded. A dysimmune profile was reported by immune-histo-chemical analysis. This interpretation might have been hindered by the prior administration of irAEs-targeted treatments.

Figure 1. Muscular / endomyocardial biopsies.

1.a. Right quadriceps muscular biopsy of case 1.

a’. Denervation: atrophic fibers & nuclear bags. b’. Diffuse expression of MHC I.

1.b. Left deltoid muscular biopsy of case 2.

a’. Perifascicular atrophy, necrosis & inflammation. b’. Diffuse expression of MHC I with perifascicular & peri-inflammatory enhancement. c’. CD3+ lymphocytes in the inflammatory infiltrate. d’. CD4+ lymphocytes in the inflammatory infiltrate. e’. CD68+ in the inflammatory infiltrate. f’. PD1+ lymphocyte expression in the inflammatory infiltrate.

1.c. Endomyocardial biopsy of case 3.

a’. Myocarditis with cardiomyocyte necrosis compatible with a toxic drug origin. b’. Mononuclear macrophages (CD68+) and lymphocytes (CD8+ and CD4+) in inflammatory infiltrates. Macrophages (CD68+) partially penetrate necrotic cardiomyocytes.

Upon the suspected diagnosis of ICI-induced myocarditis associated with myositis, the patient received urgent intravenous corticosteroid course (1mg/kg/day). Her condition secondarily deteriorated with ARF requiring IMV and rapid ventricular arrhythmia requiring emergent electrical cardioversion. Although constantly increasing troponin levels were observed with a maximum level of 1055ng/L, cardiac ultrasound close monitoring remained normal. Cardiac magnetic resonance imaging (cardiac MRI) showed signs of myocarditis with T2 myocardial signal enhancement overlying anterior and apical left ventricle wall. The coronary angiography was normal (a few parietal irregularities on left anterior descending artery <30% were considered insignificant and left untreated).
Confronted to refractory ICI-induced myocarditis associated with myositis, specific therapies were initiated with high-dose methylprednisolone (1000mg/day for 3 days, then 2 mg/kg/day), repeated plasmapheresis (seven sessions over the ICU stay) and abatacept infusions (500mg at day 0 and day 15).

The overall outcome was favorable as the patient was discharged from the hospital fifty days after the onset of irAEs. Myositis fully and rapidly resolved but ascending troponin levels concomitant to corticosteroid tapering required a slow weaning. The follow-up at six months after the irAEs diagnosis revealed an ongoing oral prednisolone (10mg/day) intake, persistently elevated troponin levels (110ng/L) with normal ECG and TTE, normal CPK levels, and a locoregional progression of the melanoma treated with targeted therapy (encorafenib / binimetinib). Nine months after ICU admission, partial response of metastatic melanoma was observed and an almost complete weaning of steroids was achieved as the patient received sole hydrocortisone 20mg/day to prevent acute adrenal insufficiency (troponin 44ng/L and unchanged TTE). A year after the myocarditis, the patient showed complete response of melanoma and troponin levels were normal even after complete weaning of corticosteroids. Figure 2.a. illustrates the chronology of events (symptoms, biomarkers, treatments).

Figure 2. Chronology of events (diagnosis, symptoms, biomarkers and treatments).

2.a. Patient 1.

CK: creatine kinase ; VF : ventricular fibrillation ; VAP : ventilator-acquired pneumonia ; DVT : deep-vein thrombosis ; *Targeted therapy : encorafenib & binimetinib

3.2. Patient 2

A 84 year-old male with a history of hypertension, asthma and metastatic melanoma diagnosed in 2019, was admitted in ICU for dyspnea associated with MG-like symptoms 19 days after the first infusion of combined anti-PD-1 (nivolumab) and anti-CTLA-4 (ipilimumab).

The oral mucosa malignant melanoma BRAFV600E- was diagnosed on the biopsy of a twenty-centimeter-large tumor of the left nasal cavity. The unresectable and metastatic (pancreas) character of the melanoma led to an ICI combination of Nivolumab + Ipilimumab as the first-line treatment upon validation par the onco-geriatric team.

Upon admission in ICU, the patient reported a symptom onset 12 days after the first infusion of ICIs. Initial general muscular weakness rapidly evolved towards a florid symptomatology associating myalgia, left then bilateral ptosis that did not fluctuate over time, binocular diplopia and ultimately, stage IV dyspnea, dysphagia and dysphonia. At ICU arrival, the patient also presented with a ubiquitous maculopapular rash. A twelve-lead electrocardiogram showed a left bundle branch block pattern and high degree atrioventricular block. According to the electro-physiologist’s advice, temporary right ventricular pacing was postponed. TTE revealed no abnormality, particularly no change in ventricular function or segmental kinetic disorder. Biological screening showed elevated troponin at 2371 ng/l (normal range 0-14) and CPK levels at 7683 UI/L, (normal range: 0-190). Inflammatory biomarkers revealed an elevated CRP at 63 mg/l, (normal range: 0-5). A baseline dNLR of 18 and an elevated NLR at 27 were assessed. Elevated liver enzymes were noted at 743UI/L and 602 UI/L for ALAT and ASAT respectively without viral or autoimmune component. Thyroid hormone levels were normal. Immunologically, complement levels were normal, along with plasma protein electrophoresis and normal immunoglobulin quantitation. Lone and unspecific speckled antinuclear antibodies at a low titer of 1/160 and a rheumatoid factor at 36 mg/l (normal range: 0-20) were encountered. Both antibodies to AChR and MuSK were absent.

A muscular biopsy of left deltoid was performed (Figure 1.b.), discarding amyloses but revealing signs of dermatomyositis with irregular and atrophic muscle fibers, some even reduced to nuclear bags. A major interstitial inflammatory infiltrate of macrophages and mononuclear lymphocytes was described along with fibers in necrosis-regeneration. Diffuse MHC I antigen expression was enhanced in perifascicular and inflammatory areas.

The patient rapidly deteriorated with ARF requiring IMV the day after ICU admission. Both cardiac MRI and coronary angiography were postponed due to respiratory and hemodynamic instability. Upon a high probability of ICI-induced acute myocarditis associated with myositis and hepatitis, the patient received urgent high-dose corticosteroid pulses (1000 mg/day for 3 days, followed by 2 mg/kg/day).

Despite the corticosteroid therapy, decreased cytolysis and CK levels, the evolution was unfavorable and the patient died before planned plasmapheresis at day 5 after admission in ICU. Figure 2.b. illustrates the chronology of events (symptoms, biomarkers, treatments).

2.b. Patient 2.

CK: creatin kinase; DM: dermatomyositis; *Targeted therapy: ipilimumab & nivolumab

3.3. Patient 3

A 70 year-old male with a history of hypertension, dyslipidemia, diabetes mellitus, and metastatic melanoma diagnosed in 2019 was admitted in ICU for an asymptomatic elevated troponin 30 days after the second infusion of combined anti-PD-1 (nivolumab), anti-CTLA-4 (ipilimumab) and anti-LAG-3 (relatlimab) (BMS CA224-048).

Stage IV superficial spreading melanoma (SSM) BRAFV600E + was diagnosed upon behavioral disorders and aphasia, revealing a cerebral lesion on the left frontal lobe. Surgical removal of the latter evidenced a melanoma metastasis. A subcutaneous nodule on the left shoulder was further identified as the primary lesion. First-line treatment provided an ICI combination (BMS048 protocol).

After the first ICI infusion, the patient presented twice in the emergency ward for febrile tachycardia attributed to ICIs at day 1 and 3 after the infusion respectively. No specific treatment was administered and the patient was discharged. At day fifteen, he received the second combination of ICIs. A fortuitous and asymptomatic elevated troponin at 280 ng/L (normal < 14) on planned blood analyzes led to the admission in cardiology two weeks after the second infusion. At arrival, the patient was still asymptomatic and the twelve-lead electrocardiogram showed sinus tachycardia and a new right bundle branch block pattern. TTE revealed lone right ventricular dilatation. Biological findings showed elevated levels of troponin (293 ng/l), CPK (2240 UI/L) and liver enzymes (120 UI/L). Although chest CT scan and coronary angiography were normal, the cardiac MRI showed signs of myocarditis involving both ventricles with late gadolinium enhancement and regional myocardial signal increase in T2 with inferior, anterior and apical distribution. The endomyocardial biopsy enhanced features of iatrogenic myocarditis with inflammatory infiltrates of mononuclear lymphocytes and macrophages along with cardiomyocyte necrosis (Figure 1.c.). Thus, high-dose methylprednisolone (1000 mg/day pulses for three days, then 2 mg/kg/day) was administered at day 19 after second ICI combination. Besides systemic corticosteroids, additional specific therapy with infliximab was administered (500mg) twenty-three days after the second ICI infusion due to ascending troponin levels (869ng/L). An intercurrent episode of high degree atrioventricular (AV) block was diagnosed at day 26 treated with the implementation of a permanent endocavitary pacemaker.

The patient was transferred in neurology at day 32 and in ICU at day 38 due to MG-like symptoms associating binocular diplopia, bilateral ptosis, proximal muscular weakness, dysphonia and a myasthenic score of 60/100. Concomitant serum levels of CPK at 818 UI/L and troponin at 695ng/L were noted. Targeted treatment included IVIG 0.4g/kg/day for five days and prostigmine. Inflammatory biomarkers were normal. A baseline dNLR of 2 and an elevated NLR at 3.6 were assessed. Thyroid hormone levels were normal. The immunologic screening ruled out a potential underlying autoimmune disease with normal complement levels, unspecific speckled antinuclear antibodies with a low titer of 1/160, absence of rheumatoid factor, anti-citrullinated peptide antibody, soluble antigen or autoimmune hepatitis-related antibodies, and normal immunoglobulin quantitation. Besides, both antibodies to acetylcholine receptor (AChR) and anti-muscle specific kinase (MuSK) were negative. Paraclinical investigations (body computed tomography scan (body CT) and ENMG) discarded neurological diseases, specifically multiple sclerosis, MG, demyelinating polyradiculoneuropathy, and other peripheral neuropathies.

His condition secondarily deteriorated with ARF requiring IMV and rapid ventricular arrhythmia requiring emergent electrical cardioversion.
Confronted to refractory ICI-induced myocarditis/myositis/MG, specific therapies were initiated with abatacept (500mg) at day 48 and repeated plasmapheresis (five sessions) from day 48 to 60. 

Despite specific treatments, improving hemodynamic and neurological status, and decreasing CK and troponin levels at 64 UI/L and 822 ng/L respectively, the patient died 62 days after the second infusion of ICI combination. Figure 2.c.. illustrates the chronology of events (symptoms, biomarkers, treatments).

2.c. Patient 3.

C1D1: first day of the first cure; CK: creatine kinase ; MRI: magnetic resonance imaging; AVB: atrioventricular block ; *Targeted therapy : nivolumab, ipilimumab & relatlimab

2.d. Comparative table of the three reported patients.

  1. Discussion

irAEs are an increasingly studied and reported entity in the oncological field27,12.Our case series highlights the four main challenges regarding irAEs encompassing:

-the lack of knowledge among physicians, thus delaying diagnosis and underestimating the severity 28

-the frequent overlapping forms of irAEs 12,29 with confusing clinical presentation rendering difficult the accurate diagnosis (Figure 3.) for emergency treatment

-and MG-like misleading presentation of myocarditis/myositis. Although the association of ICI-induced MG / myositis was described 29,only five cases of ICI-induced myositis initially presenting as misleading MG-like were reported in literature prior to our series,30.

Figure 3. Overlapping forms of severe irAEs.

irAE : immune-related adverse event ; CPK : creatin phosphokinase ; BNP : bone natriuretic peptide ; LVEF : left ventricular ejection fraction ; ECG : electrocardiogram ; MRI : magnetic resonance imaging ; PET : positron emission tomography

Thus, the diffusion of basic knowledge and irAEs’ standard of care among emergency physicians and intensivists constitutes the key to early and accurate diagnosis, proper orientation, and urgent targeted treatment as both overall prognosis and irAEs’ reversibility improve with early care31,32. Prognosis also depends on the grade of irAEs and on the association between irAEs12.

Alongside the aforementioned acknowledgement of irAEs and standardized protocols, an early assessment of risk factors for severe irAEs should be performed beforehand.33,34 Risk factors are clinical (younger age35, BMI >23kg/m² 31, current or former smoking status 36, history of type I hypersensitivity37), cancer-related35, drug-related (multiple cycles of pembrolizumab31, the use of PD1 or CTLA4 inhibitors36, ICI combination35), and biological 36.

Pathophysiology underlying irAEs likely involves multiple mechanisms2. Involvement of the lymphoid compartment with ICI-induced T cell hyperactivation is a cornerstone of the pathophysiology of irAEs. Indeed, tissue infiltration with activated T lymphocytes has been demonstrated38, notably with CD8+ T cell infiltrates in the myocardial tissue13. The muscular/endomyocardial biopsies performed in our case series also revealed significant CD3+, CD4+ and CD8+ lymphocyte infiltrates (Figures 1.a., 1.b., and 1.c.).  ICIs impact on the immunologic homeostasis by changing the T cell repertoire, thus leading to a self-reactive T cell contingent (above with antiCTLA4 use), cytokine-producing T cells39, and clonal expansion of rearranged TCR V-b sequences40. ICIs also interfere with self-tolerance by depleting regulatory T cells, thereby activating previously anergic self-reactive T cells41. Shared TCR clonality between tumor-infiltrating and myocardial-infiltrating T cells without IgG deposits may explain the occurrence of ICI-related myocarditis.12  ICIs also prevent inhibitory signal pathways leading to an overexpression of allergen-specific CD4+ T cells involved in Th2 differentiation, cytokine release (IFN-γ, TNF-α, IL-5, IL-13 or IL-17F, IL-4), and immunoglobulin class switching of B cells, thus  promoting type I hypersensitivity reactions and irAEs37. Other pathophysiological mechanisms have been hypothesized, though displaying a minor or inconsistent role in the development of irAEs (preexisting autoimmune conditions42,43 with circulating autoantibodies43,38, shared self-antigen expression by both healthy tissues and the tumor44, myeloid compartment disturbances with increased levels of circulating plasmablasts, and increased cytokine production20,38, and direct toxicity of ICIs binding to both tissues and complement38). irAEs are a double-edged sword as their occurrence is a positive predictive factor of treatment response 42,43,45 while grade 3/4 irAEs can be fatal. In regards to pathophysiological findings highlighting a major role of hyperactivated T lymphocytes, immunosuppressive therapies targeting T cells rather than antibody depletion (plasmapheresis, IVIG) seem the best option for irAEs.

Although potentially life-threatening, irAEs have been significantly associated with an ICI benefit regarding the overall response rate, the progression-free survival, and the overall survival in patients with recurrent/metastatic solid cancer46,4748. Our first case illustrates this hypothesis.

           Regarding treatment of irAEs, recent American and European guidelines recommended to stop ICIs and to urgently start high-dose corticosteroids (1-2mg/kg/day), preferably within the first 24 hours following symptoms 10, 17. Besides immediate care, case 1 also highlights the necessity of a very progressive and careful weaning over several. Indeed, a steroid tapering should be considered after 48 hours of consistent symptom improvement, and extended over 4–6 weeks12. A scrupulous and large monitoring is recommended as troponin levels alone were not systematically correlated with the risk of cardiovascular events50.

Immunosuppressive therapies (infliximab, mycophenolate mofetil, antithymocyte globulin, tacrolimus) are generally proposed for refractory and severe forms of irAEs10, 51 . Nonetheless, ICU patients diagnosed with ICI-induced myositis/myocarditis should be urgently treated with a T cell targeted therapy concurrently with steroids as it conditions the prognosis. The choice of treatment targeting irAEs is driven by the aforementioned histopathology20 and the organ-specific lesion. Abatacept seem the most accurate therapy for ICI-induced myocarditis13,52  

We proposed on Figure 4. a brief summary to promptly detect patients with irAEs in the Emergency or Intensive care wards, along with an initial screening and accurate care within the first 24 hours.

Figure 4. Simplified protocol for screening and care of irAEs in the Emergency ward or in ICU.

ICI: immune checkpoint inhibitor; GBS: Guillain-Barré syndrome; CBC: complete blood count; BNP: bone natriuretic peptide; CPK: creatin phosphokinase; CRP: C reactive protein; ANA: antinuclear antibodies; ENA: extractable nuclear antigen; ECG: electrocardiogram; TTE: transthoracic echocardiogram; TAP CT scan: thoracoabdominopelvic computed tomography scan; ENMG: electroneuromyogram; MRI: magnetic resonance imaging; SOC: standard of care; ICU: intensive care unit.

This paper has some limits as we described solely three cases. Besides, histopathology was suboptimal as late muscular biopsy was performed in case 1, possibly concealing inflammatory signs, and endomyocardial biopsy was not performed for this patient. Strengths of this article include the rarity and peculiarity of the clinical presentation, the highlights on the need for better diffusion of irAEs acknowledgment among physicians, and the long-lasting follow-up of nine months for the surviving patient.

  1. Conclusions

We described three cases of ICI-induced severe myocarditis / myositis with a misleading MG-like presentation. These cases firstly highlight the necessity to manage these patients in specified units and secondly the absolute urgency of initiating aggressive immunosuppressive treatment with high dose corticosteroids combined with anti-T cell treatment chosen upon multidisciplinary decision.

List of abbreviations.

Author Contributions:

Funding:.

Institutional Review Board Statement: I

Informed Consent Statement:

Data Availability Statement:

Conflicts of Interest:

References

  1. Postow MA, Callahan MK, Wolchok JD. Immune Checkpoint Blockade in Cancer Therapy. J Clin Oncol. 2015;33:1974–1982.
  2. Hu J-R, Florido R, Lipson EJ, Naidoo J, Ardehali R, Tocchetti CG, Lyon AR, Padera RF, Johnson DB, Moslehi J. Cardiovascular toxicities associated with immune checkpoint inhibitors. Cardiovasc Res. 2019;115:854–868.
  3. Salem J-E, Manouchehri A, Moey M, Lebrun-Vignes B, Bastarache L, Pariente A, Gobert A, Spano J-P, Balko JM, Bonaca MP, Roden DM, Johnson DB, Moslehi JJ. Cardiovascular toxicities associated with immune checkpoint inhibitors: an observational, retrospective, pharmacovigilance study. Lancet Oncol. 2018;19:1579–1589.
  4. Larkin J, Chiarion-Sileni V, Gonzalez R, Grob JJ, Cowey CL, Lao CD, Schadendorf D, Dummer R, Smylie M, Rutkowski P, Ferrucci PF, Hill A, Wagstaff J, Carlino MS, Haanen JB, Maio M, Marquez-Rodas I, McArthur GA, Ascierto PA, Long GV, Callahan MK, Postow MA, Grossmann K, Sznol M, Dreno B, Bastholt L, Yang A, Rollin LM, Horak C, Hodi FS, Wolchok JD. Combined Nivolumab and Ipilimumab or Monotherapy in Untreated Melanoma. N Engl J Med. 2015;373:23–34.
  5. Tawbi HA, Schadendorf D, Lipson EJ, Ascierto PA, Matamala L, Castillo Gutiérrez E, Rutkowski P, Gogas HJ, Lao CD, De Menezes JJ, Dalle S, Arance A, Grob J-J, Srivastava S, Abaskharoun M, Hamilton M, Keidel S, Simonsen KL, Sobiesk AM, Li B, Hodi FS, Long GV, RELATIVITY-047 Investigators. Relatlimab and Nivolumab versus Nivolumab in Untreated Advanced Melanoma. N Engl J Med. 2022;386:24–34.
  6. FDA approves anti-LAG3 checkpoint. Nat Biotechnol. 2022;40:625.
  7. Lynch TJ, Bondarenko I, Luft A, Serwatowski P, Barlesi F, Chacko R, Sebastian M, Neal J, Lu H, Cuillerot J-M, Reck M. Ipilimumab in combination with paclitaxel and carboplatin as first-line treatment in stage IIIB/IV non-small-cell lung cancer: results from a randomized, double-blind, multicenter phase II study. J Clin Oncol. 2012;30:2046–2054.
  8. Hodi FS, O’Day SJ, McDermott DF, Weber RW, Sosman JA, Haanen JB, Gonzalez R, Robert C, Schadendorf D, Hassel JC, Akerley W, van den Eertwegh AJM, Lutzky J, Lorigan P, Vaubel JM, Linette GP, Hogg D, Ottensmeier CH, Lebbé C, Peschel C, Quirt I, Clark JI, Wolchok JD, Weber JS, Tian J, Yellin MJ, Nichol GM, Hoos A, Urba WJ. Improved survival with ipilimumab in patients with metastatic melanoma. N Engl J Med. 2010;363:711–723.
  9. Motzer RJ, Escudier B, McDermott DF, George S, Hammers HJ, Srinivas S, Tykodi SS, Sosman JA, Procopio G, Plimack ER, Castellano D, Choueiri TK, Gurney H, Donskov F, Bono P, Wagstaff J, Gauler TC, Ueda T, Tomita Y, Schutz FA, Kollmannsberger C, Larkin J, Ravaud A, Simon JS, Xu L-A, Waxman IM, Sharma P, CheckMate 025 Investigators. Nivolumab versus Everolimus in Advanced Renal-Cell Carcinoma. N Engl J Med. 2015;373:1803–1813.
  10. Wang DY, Salem J-E, Cohen JV, Chandra S, Menzer C, Ye F, Zhao S, Das S, Beckermann KE, Ha L, Rathmell WK, Ancell KK, Balko JM, Bowman C, Davis EJ, Chism DD, Horn L, Long GV, Carlino MS, Lebrun-Vignes B, Eroglu Z, Hassel JC, Menzies AM, Sosman JA, Sullivan RJ, Moslehi JJ, Johnson DB. Fatal Toxic Effects Associated With Immune Checkpoint Inhibitors: A Systematic Review and Meta-analysis. JAMA Oncol. 2018;4:1721–1728.
  11. Ramos-Casals M, Brahmer JR, Callahan MK, Flores-Chávez A, Keegan N, Khamashta MA, Lambotte O, Mariette X, Prat A, Suárez-Almazor ME. Immune-related adverse events of checkpoint inhibitors. Nat Rev Dis Primers. 2020;6:38.
  12. Martins F, Sofiya L, Sykiotis GP, Lamine F, Maillard M, Fraga M, Shabafrouz K, Ribi C, Cairoli A, Guex-Crosier Y, Kuntzer T, Michielin O, Peters S, Coukos G, Spertini F, Thompson JA, Obeid M. Adverse effects of immune-checkpoint inhibitors: epidemiology, management and surveillance. Nat Rev Clin Oncol. 2019;16:563–580.
  13. Johnson DB, Balko JM, Compton ML, Chalkias S, Gorham J, Xu Y, Hicks M, Puzanov I, Alexander MR, Bloomer TL, Becker JR, Slosky DA, Phillips EJ, Pilkinton MA, Craig-Owens L, Kola N, Plautz G, Reshef DS, Deutsch JS, Deering RP, Olenchock BA, Lichtman AH, Roden DM, Seidman CE, Koralnik IJ, Seidman JG, Hoffman RD, Taube JM, Diaz LA, Anders RA, Sosman JA, Moslehi JJ. Fulminant Myocarditis with Combination Immune Checkpoint Blockade. N Engl J Med. 2016;375:1749–1755.
  14. Mahmood SS, Fradley MG, Cohen JV, Nohria A, Reynolds KL, Heinzerling LM, Sullivan RJ, Damrongwatanasuk R, Chen CL, Gupta D, Kirchberger MC, Awadalla M, Hassan MZO, Moslehi JJ, Shah SP, Ganatra S, Thavendiranathan P, Lawrence DP, Groarke JD, Neilan TG. Myocarditis in Patients Treated With Immune Checkpoint Inhibitors. J Am Coll Cardiol. 2018;71:1755–1764.
  15. Makunts T, Saunders IM, Cohen IV, Li M, Moumedjian T, Issa MA, Burkhart K, Lee P, Patel SP, Abagyan R. Myocarditis occurrence with cancer immunotherapy across indications in clinical trial and post-marketing data. Sci Rep. 2021;11:17324.
  16. Moslehi JJ, Johnson DB, Sosman JA. Myocarditis with Immune Checkpoint Blockade. N Engl J Med. 2017;376:292.
  17. Zhang L, Zlotoff DA, Awadalla M, Mahmood SS, Nohria A, Hassan MZO, Thuny F, Zubiri L, Chen CL, Sullivan RJ, Alvi RM, Rokicki A, Murphy SP, Jones-O’Connor M, Heinzerling LM, Barac A, Forrestal BJ, Yang EH, Gupta D, Kirchberger MC, Shah SP, Rizvi MA, Sahni G, Mandawat A, Mahmoudi M, Ganatra S, Ederhy S, Zatarain-Nicolas E, Groarke JD, Tocchetti CG, Lyon AR, Thavendiranathan P, Cohen JV, Reynolds KL, Fradley MG, Neilan TG. Major Adverse Cardiovascular Events and the Timing and Dose of Corticosteroids in Immune Checkpoint Inhibitor-Associated Myocarditis. Circulation. 2020;141:2031–2034.
  18. Brahmer JR, Lacchetti C, Schneider BJ, Atkins MB, Brassil KJ, Caterino JM, Chau I, Ernstoff MS, Gardner JM, Ginex P, Hallmeyer S, Holter Chakrabarty J, Leighl NB, Mammen JS, McDermott DF, Naing A, Nastoupil LJ, Phillips T, Porter LD, Puzanov I, Reichner CA, Santomasso BD, Seigel C, Spira A, Suarez-Almazor ME, Wang Y, Weber JS, Wolchok JD, Thompson JA, National Comprehensive Cancer Network. Management of Immune-Related Adverse Events in Patients Treated With Immune Checkpoint Inhibitor Therapy: American Society of Clinical Oncology Clinical Practice Guideline. J Clin Oncol. 2018;36:1714–1768.
  19. Joseph A, Simonaggio A, Stoclin A, Vieillard-Baron A, Geri G, Oudard S, Michot J-M, Lambotte O, Azoulay E, Lemiale V. Immune-related adverse events: a retrospective look into the future of oncology in the intensive care unit. Ann Intensive Care. 2020;10:143.
  20. Martins F, Sykiotis GP, Maillard M, Fraga M, Ribi C, Kuntzer T, Michielin O, Peters S, Coukos G, Spertini F, Thompson JA, Obeid M. New therapeutic perspectives to manage refractory immune checkpoint-related toxicities. The Lancet Oncology. 2019;20:e54–e64.
  21. Esfahani K, Buhlaiga N, Thébault P, Lapointe R, Johnson NA, Miller WH. Alemtuzumab for Immune-Related Myocarditis Due to PD-1 Therapy. N Engl J Med. 2019;380:2375–2376.
  22. Esfahani K, Elkrief A, Calabrese C, Lapointe R, Hudson M, Routy B, Miller WH, Calabrese L. Moving towards personalized treatments of immune-related adverse events. Nat Rev Clin Oncol. 2020;17:504–515.
  23. Martins F, Obeid M. Personalized treatment of immune-related adverse events — balance is required. Nat Rev Clin Oncol. 2020;17:517–517.
  24. Solimando AG, Crudele L, Leone P, Argentiero A, Guarascio M, Silvestris N, Vacca A, Racanelli V. Immune Checkpoint Inhibitor-Related Myositis: From Biology to Bedside. IJMS. 2020;21:3054.
  25. Toulouse E, Masseguin C, Lafont B, McGurk G, Harbonn A, A Roberts J, Granier S, Dupeyron A, Bazin JE. French legal approach to clinical research. Anaesth Crit Care Pain Med. 2018;37:607–614.
  26. Jaretzki A, Barohn RJ, Ernstoff RM, Kaminski HJ, Keesey JC, Penn AS, Sanders DB. Myasthenia gravis: recommendations for clinical research standards. Task Force of the Medical Scientific Advisory Board of the Myasthenia Gravis Foundation of America. Neurology. 2000;55:16–23.
  27. Li L, Li G, Rao B, Dong A-H, Liang W, Zhu J-X, Qin M-P, Huang W-W, Lu J-M, Li Z-F, Wu Y-Z. Landscape of immune checkpoint inhibitor-related adverse events in Chinese population. Sci Rep. 2020;10:15567.
  28. Ghisoni E, Wicky A, Bouchaab H, Imbimbo M, Delyon J, Gautron Moura B, Gérard CL, Latifyan S, Özdemir BC, Caikovski M, Pradervand S, Tavazzi E, Gatta R, Marandino L, Valabrega G, Aglietta M, Obeid M, Homicsko K, Mederos Alfonso NN, Zimmermann S, Coukos G, Peters S, Cuendet MA, Di Maio M, Michielin O. Late-onset and long-lasting immune-related adverse events from immune checkpoint-inhibitors: An overlooked aspect in immunotherapy. European Journal of Cancer. 2021;149:153–164.
  29. Pathak R, Katel A, Massarelli E, Villaflor VM, Sun V, Salgia R. Immune Checkpoint Inhibitor–Induced Myocarditis with Myositis/Myasthenia Gravis Overlap Syndrome: A Systematic Review of Cases. The Oncol. 2021;onco.13931.
  30. Valenti-Azcarate R, Esparragosa Vazquez I, Toledano Illan C, Idoate Gastearena MA, Gállego Pérez-Larraya J. Nivolumab and Ipilimumab-induced myositis and myocarditis mimicking a myasthenia gravis presentation. Neuromuscular Disorders. 2020;30:67–69.
  31. Eun Y, Kim IY, Sun J-M, Lee J, Cha H-S, Koh E-M, Kim H, Lee J. Risk factors for immune-related adverse events associated with anti-PD-1 pembrolizumab. Sci Rep. 2019;9:14039.
  32. Connolly C, Bambhania K, Naidoo J. Immune-Related Adverse Events: A Case-Based Approach. Front Oncol. 2019;9:530.
  33. Jing Y, Liu J, Ye Y, Pan L, Deng H, Wang Y, Yang Y, Diao L, Lin SH, Mills GB, Zhuang G, Xue X, Han L. Multi-omics prediction of immune-related adverse events during checkpoint immunotherapy. Nat Commun. 2020;11:4946.
  34. Michailidou D, Khaki AR, Morelli MP, Diamantopoulos L, Singh N, Grivas P. Association of blood biomarkers and autoimmunity with immune related adverse events in patients with cancer treated with immune checkpoint inhibitors. Sci Rep. 2021;11:9029.
  35. Kalinich M, Murphy W, Wongvibulsin S, Pahalyants V, Yu K-H, Lu C, Wang F, Zubiri L, Naranbhai V, Gusev A, Kwatra SG, Reynolds KL, Semenov YR. Prediction of severe immune-related adverse events requiring hospital admission in patients on immune checkpoint inhibitors: study of a population level insurance claims database from the USA. J Immunother Cancer. 2021;9:e001935.
  36. Suazo-Zepeda E, Bokern M, Vinke PC, Hiltermann TJN, de Bock GH, Sidorenkov G. Risk factors for adverse events induced by immune checkpoint inhibitors in patients with non-small-cell lung cancer: a systematic review and meta-analysis. Cancer Immunol Immunother. 2021;70:3069–3080.
  37. Shimozaki K, Sukawa Y, Sato Y, Horie S, Chida A, Tsugaru K, Togasaki K, Kawasaki K, Hirata K, Hayashi H, Hamamoto Y, Kanai T. Analysis of risk factors for immune-related adverse events in various solid tumors using real-world data. Future Oncology. 2021;17:2593–2603.
  38. Sullivan RJ, Weber JS. Immune-related toxicities of checkpoint inhibitors: mechanisms and mitigation strategies. Nat Rev Drug Discov [Internet]. 2021 [cited 2021 Nov 19];Available from: http://www.nature.com/articles/s41573-021-00259-5
  39. Postow MA, Sidlow R, Hellmann MD. Immune-Related Adverse Events Associated with Immune Checkpoint Blockade. N Engl J Med. 2018;378:158–168.
  40. Robert L, Tsoi J, Wang X, Emerson R, Homet B, Chodon T, Mok S, Huang RR, Cochran AJ, Comin-Anduix B, Koya RC, Graeber TG, Robins H, Ribas A. CTLA4 Blockade Broadens the Peripheral T-Cell Receptor Repertoire. Clin Cancer Res. 2014;20:2424–2432.
  41. von Itzstein MS, Khan S, Gerber DE. Investigational Biomarkers for Checkpoint Inhibitor Immune-Related Adverse Event Prediction and Diagnosis. Clinical Chemistry. 2020;66:779–793.
  42. Weinmann SC, Pisetsky DS. Mechanisms of immune-related adverse events during the treatment of cancer with immune checkpoint inhibitors. Rheumatology. 2019;58:vii59–vii67.
  43. Yoest J. Clinical features, predictive correlates, and pathophysiology of immune-related adverse events in immune checkpoint inhibitor treatments in cancer: a short review. ITT. 2017;Volume 6:73–82.
  44. Berner F, Bomze D, Diem S, Ali OH, Fässler M, Ring S, Niederer R, Ackermann CJ, Baumgaertner P, Pikor N, Cruz CG, van de Veen W, Akdis M, Nikolaev S, Läubli H, Zippelius A, Hartmann F, Cheng H-W, Hönger G, Recher M, Goldman J, Cozzio A, Früh M, Neefjes J, Driessen C, Ludewig B, Hegazy AN, Jochum W, Speiser DE, Flatz L. Association of Checkpoint Inhibitor–Induced Toxic Effects With Shared Cancer and Tissue Antigens in Non–Small Cell Lung Cancer. JAMA Oncol. 2019;5:1043.
  45. Sanlorenzo M, Vujic I, Daud A, Algazi A, Gubens M, Luna SA, Lin K, Quaglino P, Rappersberger K, Ortiz-Urda S. Pembrolizumab Cutaneous Adverse Events and Their Association With Disease Progression. JAMA Dermatol. 2015;151:1206.
  46. Das S, Johnson DB. Immune-related adverse events and anti-tumor efficacy of immune checkpoint inhibitors. J Immunother Cancer. 2019;7:306.
  47. Masuda K, Shoji H, Nagashima K, Yamamoto S, Ishikawa M, Imazeki H, Aoki M, Miyamoto T, Hirano H, Honma Y, Iwasa S, Okita N, Takashima A, Kato K, Boku N. Correlation between immune-related adverse events and prognosis in patients with gastric cancer treated with nivolumab. BMC Cancer. 2019;19:974.
  48. Foster CC, Couey MA, Kochanny SE, Khattri A, Acharya RK, Tan Y-HC, Brisson RJ, Leidner RS, Seiwert TY. Immune-related adverse events are associated with improved response, progression-free survival, and overall survival for patients with head and neck cancer receiving immune checkpoint inhibitors. Cancer. 2021;127:4565–4573.
  49. Curigliano G, Lenihan D, Fradley M, Ganatra S, Barac A, Blaes A, Herrmann J, Porter C, Lyon AR, Lancellotti P, Patel A, DeCara J, Mitchell J, Harrison E, Moslehi J, Witteles R, Calabro MG, Orecchia R, de Azambuja E, Zamorano JL, Krone R, Iakobishvili Z, Carver J, Armenian S, Ky B, Cardinale D, Cipolla CM, Dent S, Jordan K, ESMO Guidelines Committee. Electronic address: clinicalguidelines@esmo.org. Management of cardiac disease in cancer patients throughout oncological treatment: ESMO consensus recommendations. Ann Oncol. 2020;31:171–190.
  50. Spallarossa P, Tini G, Sarocchi M, Arboscello E, Grossi F, Queirolo P, Zoppoli G, Ameri P. Identification and Management of Immune Checkpoint Inhibitor-Related Myocarditis: Use Troponin Wisely. J Clin Oncol. 2019;37:2201–2205.
  51. Cautela J, Zeriouh S, Gaubert M, Bonello L, Laine M, Peyrol M, Paganelli F, Lalevee N, Barlesi F, Thuny F. Intensified immunosuppressive therapy in patients with immune checkpoint inhibitor-induced myocarditis. J Immunother Cancer. 2020;8:e001887.
  52. Salem J-E, Allenbach Y, Vozy A, Brechot N, Johnson DB, Moslehi JJ, Kerneis M. Abatacept for Severe Immune Checkpoint Inhibitor-Associated Myocarditis. N Engl J Med. 2019;380:2377–2379.
